# Green Building in the Arctic Region: State-of-the-Art and Future Research Opportunities

**Lucrezia Ravasio \*, Svein-Erik Sveen and Raymond Riise**

Department of Building, Energy and Material Technology, UiT—The Arctic University of Norway, 8515 Narvik, Norway; svein-erik.sveen@uit.no (S.-E.S.); raymond.riise@uit.no (R.R.)
\* Correspondence: l.ravasio@uit.no; Tel.: +47-93959483

**Abstract:** The concept of Green Building refers to environmentally friendly constructions with the target of minimizing the impact on the natural environment through sustainable and efficient use of resources over their life cycle. Since modern buildings are large contributors to global energy consumption and greenhouse gas emissions, policies and international strategies intended to reduce the carbon footprint of conventional buildings are highlighting the role of this recently introduced building concept. This study provides a systematic literature review of existing research related to Green Buildings in the Arctic. Despite numerous studies and projects developed during the last decades, a study describing the current research status for this region is still missing. The review first examines the role that national and international policies developed by the arctic countries have on the development process of Green Buildings. Second, it provides an overview of the most commonly used and promoted Green Building rating systems used by the same countries in the region. The analysis highlights benefits and critical issues of Green Buildings located in the Arctic in comparison with conventional buildings, focusing on environmental, economic, and social dimensions. Finally, future research opportunities are presented and discussed.

**Keywords:** Green Building; arctic; literature review; sustainability

---

## 1. Introduction

In recent decades, the consciousness of the impact of human activity on the natural environment has grown. This awareness has affected the construction industry, highlighting the link between *sustainability* and *environment* and thereby giving it strength and momentum [1]. The green movement, having spread in all fields of society, has led to the emergence of worldwide, national, and local programs advancing green principles in both construction and home-building sectors [2]. Indeed, studies show that buildings play a significant role in climate change.

The term *climate change* generally refers to the long-term shift in global or local climate patterns, usually identified with the rise of average temperature over the years, owing to human activities. Among all the regions of the planet, because of its special physical and geographical properties, the Arctic is experiencing the most severe effect of climate change through greater and more rapid rise of average temperature [3]. The *Arctic Region* is commonly defined as the area north of the Arctic Circle (66°32′ N), or as the area north of the 10 °C July isotherm as shown in Figure 1a. Alternatively, it can also be defined by vegetation or oceanographic characteristics. In this review, the definition used by the Arctic Monitoring and Assessment Program (AMAP) is adopted [4]. It considers the area delimited by the tree line as shown in Figure 1b. The arctic climate is typically characterized by extreme seasonality and variation in temperature and precipitation, strong gradient in latitude solar, and UV radiation [5]. In addition, low temperatures lead to an extensive and permanently ice-covered or frozen ground, i.e., *permafrost*, which makes the region vulnerable to climate change. Warming of the

Arctic and consequent melting has global implications, such as alteration of global ocean circulation, sea level rise, and release of methane and carbon dioxide trapped in the permafrost, i.e., gases that are feeding and accelerating the process of temperature-rise [6].

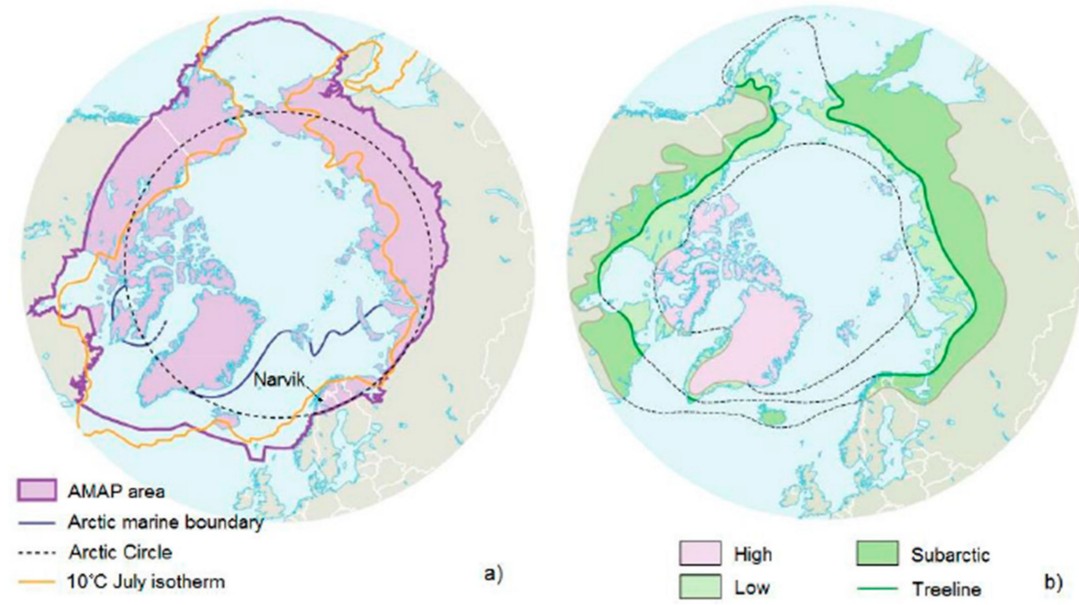

**Figure 1.** (**a**) The Arctic defined by the 10 °C July isotherm. (**b**) Arctic floristic boundaries [7].

According to the Global Status Report of 2019 [8], buildings and constructions together account for 36% of global energy use and 39% of energy-related carbon dioxide emissions in 2018. This makes buildings the largest contributing sector to global warming. The same report declares that, due to the strong floor area and population expansions, total global energy consumption in buildings in 2018 increased 1% from 2017. In perspective, Green Buildings become a potential strategy and investment to limit demand and reduce energy intensity. In fact, through the introduction of new building codes and adoption of advanced certifications for high-energy performance, all participants in the building and construction value chain are globally contributing to decarbonization of building stocks and to the improvement of building's energy performances.

By definition, a *Green Building* is a high-performance building with a reduced negative impact on the natural environment and human health [9]. This is achieved by applying measures that take into account the building location, as well as water, energy and material use efficiency, resource conservation, indoor air quality, building operation, and maintenance over the entire building life-cycle [10]. Green Buildings also provide benefits from an economic and social perspective, through lower building life-cycle costs and improved comfort and well-being of their occupants [11]. This promising solution is also expressed in different building concepts related to sustainable and environmental design such as net and nearly zero-energy buildings, zero-emission, zero-carbon, and carbon-neutral buildings [12].

With this backdrop, policies aimed at safeguarding and protecting the arctic environment represent a challenge of paramount importance for the region at the present and for the future [13]. Governments with territories in the Arctic—Norway, Sweden, Finland, Denmark, Iceland, Russia, Canada, and United States—are closely involved in the development of new initiatives both locally, with national legislation, and globally, through the *Arctic Council*. Established in 1996 with the Ottawa Declaration, the Arctic Council is an intergovernmental forum promoting cooperation, coordination and interaction among the Arctic States [14]. The Arctic Cooperation also includes the European Union, the Nordic Cooperation, the Barents Cooperation, and the United Nations [15].

The purpose of this research is to present local and global initiatives stated by the institutions and the bodies on reduction of building's carbon footprint in the Arctic. The aim is to examine the players in the Green Buildings transition process in the Arctic and evaluate the applicability of currently used assessment tools in these special climate conditions. By identifying benefits and criticalities, and design practices of Green Buildings Arctic climate, the study highlights the current state of the art and future research opportunities. Therefore, this results in practical interest for companies and institutions that want to invest resources for development solutions for reducing the buildings' carbon footprint in the region.

## 2. Methodology

In order to provide an overview of the status of sustainable constructions in the Arctic, the scope and disposition of the paper are in accordance with the standards of a systematic literature review. The method chosen consists of several steps, aimed at identifying, analyzing, and interpreting the literature available from a specific period in time, related to the research subject [16].

The review process was initiated by identifying questions related to the topic of the study. The aim was to delineate the direction that the review has to follow, the questions it has to answer, and the added value that it could provide to the scientific and academic community. Secondly, relevant databases for the research were utilized. Scopus, MDPI, Spring, and High North Research Documents—a database providing literature within the thematic scope of the circumpolar high north—fulfilling the objectives of the research. Due to the young age of the research topic, no time boundaries were imposed on the literature search. For instance, referred to the literature range from 1997 to 2020, and written in English. It includes books, journal papers, and conference papers for the discussion of paragraph 1 and Sections 3.2 and 3.3. Information related to Sections 3.1 and 3.2—such as reports and legislation acts—was extracted by the official websites of the governments and organizations cited in the study. In order to find material centering on Green Buildings in the Arctic, keywords definitions and the corresponding synonyms used in the databases were identified. Results were skimmed, first eliminating duplicates, and then applying quality criteria for identifying articles and books integrating 'green buildings' or 'arctic' in their title, abstract, or keywords. The articles referred to were classified and grouped into (1) introductory background on sustainability of buildings; (2) arctic and climate change; (3) construction policies of the arctic countries; (4) rating systems for Green Building; and (5) benefits and criticalities of green buildings.

The following paragraphs present the results, highlighting gaps and limitations both in research and in literature, and the future research opportunities.

## 3. Results

### 3.1. Sustainable Buildings and Construction Policies

Worldwide, climate change actions and policies are primarily driven by the Paris Agreement. Established in 2015, the Paris Agreement, signed by 190 countries, sets out a global framework of targets to [17]:

1.   Keep the increase in global average temperature to well below 2 °C by the end of the century, preferably limit the temperature increase to just 1.5 °C;
2.   Increase the ability to adapt to climate change;
3.   Make finance flows consistent with a pathway towards low greenhouse gas emissions and climate-resilience development.

According to the Paris Agreement, 2020 is a key-year. Countries are asked to communicate their new or updated nationally determined contributions (NDCs) delineating their efforts to reduce national emissions, and adapt to the impacts of climate change. In view of this challenging target, and the effects of global warming in the Arctic as summarized in the previous section, this paragraph

introduces medium- and long-term strategies developed by the arctic governments for the housing and building sector of the region. The aim is to analyze the relationship between policies for mitigation of climate change in the Arctic and policies for decarbonization and improvement of energy performance of buildings at a national level, thus highlighting the role such legislation has on the Green Building growth-process.

The following subsections present a brief synthesis of national strategies for the Arctic and for the building sector for the following Arctic Countries: Norway, Sweden, Finland, Iceland, Russia, Canada, and United States with Alaska. Denmark, whose territories of Greenland and Faroe Islands are part of the Arctic, has been excluded from the discussion because, according to the Arctic Human Development Report of 2003 [18], building emissions are not considered to significantly affect climate conditions due to the small population size. Data relating to national strategies have been extracted from the official publication channels of the respective governments.

### 3.1.1. Norway

The strategy for fulfilling targets of the Paris Agreement is presented in the "Climate Change Act" released in 2018 [19]. The national goals to achieve together with the European Union (EU) include reduction of greenhouse gases by at least 40% by 2030 and reducing of greenhouse gas emissions in the order of 80–95% by 2050, resulting in Norway becoming a low-emission society. In both cases, the reference year is 1990 and climate targets should be reviewed every five years. The Climate Change Act identifies five priorities areas: Transportation sector; supply of renewable energies; low emissions and clean production technologies; environmentally sound shipping; and carbon capture and storage.

Development in the Arctic has also been a priority in the Norwegian Government's agenda since 2005, demonstrated by several proposals released over the years. The most significant official publications include "New Building Blocks for the High North" and "Norway's Arctic Strategies between geopolitics and social development". The first program, established in 2006 and released in 2009, contains 22 specific action points enclosed in seven prioritized areas ranging from technical to humanity. The purpose of the project is to enhance knowledge in and about the north, increasing government activity and presence in the area, and lay foundations for sustainable economic and social development in the Arctic regions [20]. The second program, presented in 2017, reveals the government's vision for economic, environmental, and social sustainability in the Arctic, highlighting the need to reduce greenhouse gas emissions and pollution through promotion and transition to green transport, energy, and construction [21]. However, both plans, released in a unified manner on a national level, lack strategies strictly related to the building sector.

As described in the document "The Property Sector's Roadmap toward 2050"—released in June 2016 by the Norwegian Green Building Council, Grønn Byggallianse, and Norsk Eiendom—the vision for 2050 is to achieve a climate-neutral construction industry with zero emissions of environmental toxins, in accordance with the Paris Agreement [22]. The requirements to meet national and international goals for greenhouse gases emissions reduction for the building sector are provided in the national building code "Regulations on technical requirements for construction works—TEK 17", whose latest version was released in July 2017 [23]. However, attention is not only on new buildings, in which energy requirements have been tightened to nearly-zero Energy Building standards in 2020 [24], but also on existing building stocks, whose performance must be upgraded in case of a planned renovation. In reference to refurbishing of existing buildings, Grønn Byggalliance released a booklet in November 2019, "Think twice before demolishing", encouraging renovation of dwellings instead of their demolishing, promoting their conversion into Green Buildings to achieve the 2050 climate goals [25]. Long-term initiatives for reducing the carbon footprint of buildings can also be found in the program "Building for the future—environmental action plan for the housing and building sector 2009–2012". It states long-term initiatives for reducing the carbon footprint of buildings, acting on their energy needs and waste production [26].

### 3.1.2. Sweden

The Swedish climate change and energy policies framework was published by the Swedish parliament in June 2018. According to EU regulations and the Paris Agreement, main objectives are set at 10-year intervals and mainly include reduction of emissions, taking 1990 as a reference year. These policies are presented in the report "Sweden's draft integrated national energy and climate plan" [27], which introduces the following measures for achieve the climate target set for 2030 for dwelling houses:

1. Limits in the specific energy use (kWh/m$^2$ and year), average thermal transmittance [W/(m$^2$K)], and building's average air leakage [1/(sm$^2$)] for new and existing buildings.
2. A support scheme for renovation and energy efficiency of rental apartments, introduced to incentivize renovation and energy efficiency of rental apartments in areas with socioeconomic challenges.
3. Establishment of an Information Centre for Sustainable Building for promoting energy-efficient renovation and energy-efficient construction using sustainable materials and low climate impact from a life-cycle perspective.
4. Implementation of an Energy Performance Certificate Act, a law on energy performance certificates for buildings, to promote efficient use of energy and healthy indoor environment.

These strategies, which are similar to those implemented in Norway, with TEK 17 and *Energi-forskriften* (Energy Regulation), were also presented in the document "Sweden's Seventh National Communication on Climate Change" [28], along with the report of the downward trend in emissions between 1990 and 2015, due to the transition from oil-fueled heating of homes and commercial to electricity.

However, actions are being taken by the government at a regional and local level, and include a new energy labelling directive (Ecodesign Act SFS 2008:112) as well as requirements for setting minimum energy performance standards (Energy Performance of Building Directive 2010/31/EC), and the implementation of a law on energy performance certificates for buildings (Energy Performance Certificate Act SFS 2006:985) [28]. Specific policies regarding the construction sector have been developed by the Swedish National Board of Housing, Building, and Planning and include the Planning and Building Act (2010:900), and the Planning and Building Ordinance (2011:338). In particular, the second chapter of the legislation 2010:900 aims to promote a planning with regards to natural and cultural values, environmental and climate aspects through also a long-lasting and effective management of land and water areas, energy resources and raw materials [29].

The strategy developed by the Swedish Government for the Arctic was presented in 2014 through the "Sweden's strategy for the Arctic region" program, where priorities and the outlook for Sweden's arctic policy have been outlined. The government's goal is to promote sustainable development in an economic, social and environmental dimension, and to reduce global emissions of greenhouse gases and short-lived climate forces, along with the implementation of the Arctic cooperation program [15].

### 3.1.3. Finland

In October 2012, the Finnish Government adopted their latest artic policy, extensively summarized in the report "Finland Strategies for the Arctic Region 2013". Once again, the main objectives are related to the promotion of stability, national and international cooperation, and sustainable development [30]. The program also examines possibilities to promote and achieve them, but a specific action for reviewing and redefining the role of buildings for the Arctic is not covered. The government's plan for the building sector is explored in the document "Government Action Plan 2017–2019" [31], where mid-term national objectives and activities for different sectors—such as Employment, Education, Health, Bioeconomy, and Digitalization—are presented through five strategic priorities and 26 key projects. Priority number 4—*Bioeconomy and Clean Solution*—reveals Finland's interest in introducing and exporting of sustainable solutions to achieve climate objectives of reducing greenhouse gases and the economical state of the

country in the Baltic Sea [31]. This general statement includes also the building sector, whose priorities are identified in the "Energy and Climate Roadmap 2050", a strategic level guide to permit attaining Finland's long-term objective of a carbon-neutral society [32]. Concerning buildings and constructions, the program outlines the necessity, in line with the Paris Agreement, of new buildings to meet nearly-zero energy standards by the end of 2020. It also emphasizes the necessity of meeting stricter energy efficiency requirements as set out by the updated national building code of 2013 for renovation, or retrofit, construction projects [32].

### 3.1.4. Iceland

Iceland's Climate Policy is introduced in the report "Iceland's Climate Action Plan for 2018–2030", released in September 2018 [33]. Once again, efforts are directed at cutting net emissions to meet the Paris Agreement targets for 2030 and reach the government's ambitious aim of carbon neutrality before 2040. The plan consists of 34 actions, divided in four categories—clean energy transfer in transport; clean energy transformation in other sectors; climate mitigation in land use and forestry; other measures—in which buildings and use of energy do not find a direct collocation. In fact, as largely covered by the document "Iceland's Seventh National Communication and Third Biennial Report", the construction sector, with a high energy-demanding space heating, accounts for the 6% of the total GHG emissions in the energy sector in 2015 [34]. However, according to the same report, 99% of energy used for space heating is already produced by renewable energy sources such as hydropower and geothermal power. Specific legislation and regulation on construction are mostly intended to ensure safety human life and the environment. Sustainable development is also a guiding concern in design and construction of energy efficiency in building operations [34].

In addition, the Arctic Council Chairmanship program 2019–2021—"Together Towards a Sustainable Arctic"—which highlight the national commitment for the sustainable development and protection of the Arctic environment, does not refer directly to a plan for the building sector [35]. Measures primarily involve the arctic marine environment, the Arctic Council, the people and the community, the climate, and green energy solutions. In this last section, the government encourages the development and application of practical green energy solutions in the Arctic to reduce emissions and improve air quality.

### 3.1.5. Russia

In 2008, the Russian Federation defined a state policy comprehended the national interest for the Arctic to be achieved by the end of 2020—"Basic Principle of Russian Federation State Policy in the Arctic to 2020". Primary goals include promotion of social and economic development, peace and cooperation, protection of the ecosystem, and a shipping route through the Northeast Passage [15]. In March 2020, the government released a new version—"Basic Principles of Russian Federation State Policy in the Arctic to 2035"—updating the goals to achieve by the end of 2035 [36]. Even though the strategy lacks direct or indirect measures for reducing the carbon footprint of buildings in the Arctic, national building legislation is continuously evolving. Indeed, to meet EE (Energy Efficiency) standards, the Government implemented rules for determining energy efficiency class of apartment buildings (Order if the Ministry of Russia n.339/pr of 6 June 2016), and the requirements for energy efficiency of building, structures, and facilities (Order if the Ministry of Russia n.1550/pr of 17 November 2017). In 2016, it also released a "Road Map for EE buildings and structures" (Russian Federation Government Order N.1853-R of 1 September 2016), in which primary objectives for the housing sector are emphasized, such as the rational use of energy resources; increase of high-energy efficiency in design and construction of buildings; and development of technical regulation and standardization in EE. In addition to new energy efficiency standards, in 2017 the government set several mandatory technical requirements regarding measuring energy consumption in new dwellings and the implementation of requirements for building envelopes. Russia's most recent plan for the building sector aims at modernizing building and production, and increasing the contribution from the technological sector in

reducing the energy consumption for the Gross National Product (GDP) by at least 1.5% per year [37]. It also aims to provide a large-scale increase of energy efficiency of Russian economy by intensifying the renewable-sources-based energy generation, as well as large-scale electrification. The goals to be achieved in order to meet the Paris Agreement's target are reduction of Russian GDP by 9% by 2030 and carbon neutrality by 2050.

### 3.1.6. Canada

The latest version of the Canadian Government's arctic policy "Arctic and Northern Policy Framework" was released in September 2019. The three key opportunities highlighted in the strategy for the circumpolar Arctic region are: Strengthening the rule-based international order in the Arctic; defining Canada's Arctic boundaries; and finally, broadening Canada's international engagement and contribution to the priorities of Canada's Arctic and North [38].

Even though Canada's arctic policy is mostly focused on international cooperation and on local communities, the government is committed to climate action policy directed at the building sector. Canada's strategy for combatting climate change considers the emissions-productive sources. Among them, homes and buildings account for 11% of Canada's total emissions. The government's long-term solution aims to create a low-carbon building sector, ensuring high-quality standards through the development of new building codes. The first is a "net-zero energy ready" model building code for new buildings. The second is a model code for existing buildings to guide the process of retrofitting buildings to accommodate energy efficiency improvements during renovations [39]. Moreover, the government aims to support home and building retrofit programs across Canada, and improve energy efficiency of historical buildings as well as building located in indigenous communities' [40].

The most recent commitment dates from March 2020, when the Canada Green Building Council also launched an initiative—financed by the *Environment and Climate Change Canada*—for designing the updated Zero Carbon Building Standards [41]. The second version (V2) of the Zero Carbon Building Standards provides more rigor to ensure zero emission and flexibility to encourage the boundless adoption of zero-carbon buildings [42].

### 3.1.7. Alaska

United States' policy for the Arctic—"National Strategy for the Arctic Region"—was first released in May 2013, implemented in January 2014, and later updated in March 2016 [43,44]. The main points covered by the strategy concern the advancing of United States security interests; pursuing responsible Arctic Region stewardship; and strengthening international cooperation. In addition, development of renewable energy resources and the adoption of sustainable strategies are on the Government's Arctic agenda.

Despite the USA government withdraw from Paris Agreement in November 2019, the main targets of Alaska for Climate Change, presented in the report "Climate Change Action Plan Recommendations to the Governor", are still to support and incentivize energy efficiency, renewable energy, decarbonization and beneficial electrification across all sectors [45]. In line with this program, the State of Alaska is currently following the Building Energy Efficiency Standards (BEES), regulations established and updated since 1991, comprised of the 2018 International Energy Conservation Code (IECC); ASHRAE 62.2 2016 and Alaska Specific Amendments. These standards aim to promote construction of energy efficient building and nowadays a minimum energy rating of five stars is required [46].

### 3.2. Green Building Rating Systems

A building is rated *green* if it satisfies a set of energy performance targets. The Green Building Council is a worldwide organization founded in 1993, working in the building and construction industry, with the mission of promoting sustainability in this sector. Nowadays, the organization counts 70 Green Building Councils around the world, that, over the years, have developed and administered many of the assessment tools aimed at evaluating and identifying buildings that meet green standards

and performance requirements [47]. By encouraging and rewarding companies and organizations operating in a green mind-set, these rating systems have become powerful tools that are transforming and pushing the boundaries of sustainability in the building sector. Indeed, they are setting standards that affect and evolve both the building codes and building-related government legislation [48].

Assessment tools can be applied to different types of constructions (e.g., residential or commercial buildings or whole neighborhoods), during different life-cycle stages (e.g., planning and design, construction, operation and maintenance, renovation or demolition), using different approaches. All rating systems have a broadly similar structure. They are typically divided into categories covering various aspects of sustainability, to which it is possible to assign a certain value or number of credits. Each category has a different weighted contribution to the overall score.

However, despite similarities, governments and organizations have developed and suggested the use of systems that comply with local climate conditions, legislation, and needs [49]. The following provides an overview of well recognized rating tools.

### 3.2.1. LEED and BREEAM

Currently, worldwide, the leading Green Building assessment tools are Leadership in Energy and Environmental Design (LEED) and Building Research Establishment's Environmental Assessment Method (BREEAM).

LEED is an American Green Building rating tool released by the U.S. Green Building Council in 1998. It offers certifications for different types of projects, such as New Construction (LEED-NC), Core and Shell (LEED-CS), Commercial Interiors (LEED-CI), and Existing Buildings (LEED-EB), that makes it versatile and capable of reaching a wide audience. In the assessment process, seven parameters are evaluated: Sustainable Sites; Water Efficiency; Energy and Atmosphere; Materials and Resource; Indoor Environmental Quality; Innovation in Design and Regional Priorities. These categories have a maximum achievable number of points and from one to three prerequisites. The base score is 100, to which 6 and 4 points are added for the Innovation and Design and Regional Priority categories.

According to the score achieved, the ranking is divided in four levels: Certified (40–49 points); Silver (50–59 points); Gold (60–79 points); and Platinum (80–110 points) [50]. LEED is currently at its fourth version.

BREEAM is a rating tool developed by Building Research Establishment in UK, launched in 1990. It assesses the environmental impact of newly constructed buildings at the Design Stage (DS) or at the Post Construction Stage (PCS). It is usually divided in 10 sections; Management; Health and Wellbeing; Water; Materials; Energy; Waste; Transport; Land Use; and Ecology; Innovation; Pollution; with an associated score and weight depending on the country being considered. BREEAM also set minimum performance standard in key areas. Based on the number of credits achieved, the final score is calculated and rated in five levels: Pass (≥30%); Good (≥45%); Very Good (≥55%); Excellent; (≥70%); and Outstanding (≥85%) [51].

### 3.2.2. Other Green Building Assessment Tools

Besides LEED and BREEAM, arctic governments and organizations are currently certifying buildings using other tools, which take into account specific local climate conditions, economic development level, and geographical characteristics.

BREEAM-NOR and BREEAM-SE are, respectively, the Norwegian and the Swedish versions of the certification system. The evaluation is performed according to the same criteria, but with different associated weighting values [52,53]. However, in Sweden, Miljöbyggnad is the leading environmental certification system, since it is based on Swedish building regulations and regulatory requirements. It is used to certify new constructions, refurbished buildings or existing buildings through the evaluation of four areas: Energy; Indoor Environment; Building Material; and Special Environmental Requirements. It has four rating levels: Rated; Bronze; Silver; and Gold [54]. In Iceland, the Green Building Council—Grænnibyggd—is mostly encouraging the use of the international version

of BREEAM and LEED, since at the moment they have not developed a version based on the local climatic and economic characteristics [55]. In Finland, the new RTS environmental classification system (RTS GLT) has also been designed in respect to Finnish conditions, legislation, and diversity of the country's building stock. It is based on European Standards (CEN TC 350 standards), together with common best practices in the sector. It evaluates five main areas: Process; Finances; Environment and Energy; Indoor Air and Health; Innovations. The final ranking is given in stars and determined by the total score achieved: 1 star (≥25 points); 2 stars (≥40 points); 3 stars (≥55 points); 4 stars (≥70 points); 5 stars (≥85 points) [56].

Information regarding Russian rating systems are not easily traceable, since most of the information is in Russian. However, the review has found that in addition to BREEAM and LEED, Green Buildings are commonly certificated through GOST R, Green Standards Certification System, or Green Zoom. GOST R is a voluntary national quality standard for construction, that includes several features of a certification system. It is based on requirements on environmental performance provided by Russian legislation and the national building code. It differs to other approaches by not assigning a final ranking or award. The Green Standards Certification System also uses the existing Russian building code and legislation as reference. The evaluation considers eight categories of different weight and a final score of 100, according to the following ranking: Certified (40–49%); silver (50–59%); gold (60–79%); platinum (80–89%). Green Zoom is the most recently introduced Russian certification system. It is a LEED-based system evaluating 48 criteria, divided into nine categories, eight of which deal with the general performance of the building, and one with local climatic regional issues [57]. Literature has not been found that provides information on the classification of the final score.

Unlike European Countries, BREEAM is not a popular rating tool in Canada and in United States. In Canada, LEED, LEED for Homes, and LEED Canada, a version meeting Canadian legislation and performance requirements, are the most widespread certifications. Along with these tools, the Building Owners and Manager Association Building Environmental Standards (BOMA best) releases five levels of certifications: Certified (≥19%); Bronze (≥20%); Silver (≥50%); Gold (≥80%); Platinum (≥90%); according to the following six areas: Energy, water, waste reduction and site, emissions and effluents, indoor environment, and environmental management system [58]. In Alaska, LEED and all its versions is the most used system for releasing certifications. The list of other popular tools also includes: Energy Star; Green Globes; Living Building Challenge; National Green Building Standard and WELL Building Standards.

### 3.3. Benefits and Criticalities of Green Building

In literature, a large number of studies related to Green Buildings have been comparing pros and cons associated with the development of green constructions, with those of conventional buildings.

Benefits are typically classified in three main categories: environmental; economic; and social. Each of these are briefly discussed below. The novelty of this paragraph is highlighting benefits and criticalities of Green Buildings in the Arctic.

### 3.3.1. Environmental Benefits

From an environmental perspective, Green Buildings help preserve the ecosystem through a conscious and sustainable use of resources. This practice involves: Reduced use of energy; reduced use of water; land conservation; material conservation.

Energy and water savings are possible utilizing energy efficiency and water saving appliances. In fact, use of natural light and ventilation, integration with green energies—such as solar, wind, or geothermal, and optimization of the effectiveness of external building envelope, lead to fuel saving and therefore reductions in emissions of pollutant gases [59].

Material and land conservation are achieved by optimizing the use of land and a rational use of building materials, leading to pollution and waste reduction through reuse and recycling [60]. These benefits are also the result of requirements given by Green Building Assessment Tools regarding

energy, waste, and water consumption. Energy efficiency not only leads to higher performance compared to conventional buildings [9], but also to significant reductions of greenhouse gas emissions and other harmful air pollutants, whose release is associated with combustion of fossil fuels for electricity and heat production [61].

### 3.3.2. Economic Benefits

Economic benefits are typically cost savings due to lower energy demand, thereby also lowering operation and maintenance costs. In fact, on average, Green Buildings use 30% less energy than conventional buildings, owing to reduced electricity usage, as well as reductions in peak energy demand [62]. Although, meeting the requirements set out in Green Buildings standards require extra costs associated with construction materials, energy-saving technologies, and the certification process. However, studies have shown that the investment is profitable given the energy savings and lower maintenance costs [63].

Technological innovations have played a key role in achieving these objectives and accreditations. Attaining the technical energy performance requirements for a building necessary affects the choice of thermal insulation and energy generation systems. By ensuring optimal choice of insulation through designing of an advanced building envelope, energy losses are limited, contributing to a stable building performance. Integration and utilization of renewable energy systems for energy generation also reduces energy consumption and emissions [64]. Despite the lack of infrastructure, connecting electricity generated at the building site to the power grid is a common challenge when launching these technologies [49]. There are however, several possibilities for exploiting new renewable energy resources in the Arctic. Norway, Sweden, and Finland have already adapted its grid for electricity produced by hydroelectric power plants, whereas Russia is running projects for energy generation, involving installation of photovoltaic panels and energy storage equipment in remote off-grid communities. Other innovative solutions include geothermal power and glacial meltwater power plants [65]. Especially in high energy-demand regions, such as the Arctic, utilizing energy-saving measures to reduce consumption and costs during the life cycle of the building is a critical and fundamental aspect that cannot be overlooked.

### 3.3.3. Social Benefits

Providing a high level of Indoor Environment Quality (IEQ) for Green Buildings is what ensures an occupant's improved health and productivity, as reflected in a higher level of comfort and performance in Green Buildings as compared to what is achieved in conventional buildings. This is achieved by the integration of a mechanical ventilation system, controlling airflow and air quality, minimizing sources of air pollution, and keeping temperatures at a comfortable level. Interior lighting quality and building acoustics contribute to the well-being of occupants [66]. However, studies have shown that difficulties in controlling temperature, ventilation, and lighting often reduce the level of user satisfaction [67–69], due to the lack of individual control.

Life cycle assessment (LCA) is another useful approach for analyzing and bringing improvements to the technical aspects of Green Buildings. LCA focuses on many aspects, from manufacturing and transportation of materials, energy and water consumptions to GHG emission during the operation stage. Through a correct analysis, LCA evaluates the impact of an entire building or a single component at an early stage, thus improving building design [70].

In the Arctic, where winters are long and dark, and people spend most of the time in doors, this category is of even further importance. Therefore, it is fundamental to have buildings improving the well-being of users as well as maintaining the flexibility of controlling systems in order to adapt to the need of users.

## 4. Discussion

In this section, the results presented in the previous paragraphs are discussed. The discussion is divided in subparagraphs, corresponding to each of the themes previously introduced.

### 4.1. Role of Policies in the Green Buildings Development

The transition from conventional buildings to Green Buildings is a process involving actors on niches, regimes, and landscape levels. The interaction of these three factors makes innovation possible in any sector. Niches are places where technologies are developed, without market pressures, while regimes are defined as the set of structures aimed at establishing practices and constitutional arrangements. Lastly, landscapes delineate factors influencing the niche-regime interaction, such as global political events and global markets. By identifying these roles in the Green Arctic Buildings development process, the review evaluates the state-of-the-art of the progression from conventional to sustainable buildings establishment.

Since regimes are the essence of transition, Section 3.1 identifies policies and regulation changes for the selected countries over the last decades. In fact, policies impacting on energy performance of buildings makes it possible to implement innovative projects developed in R&D departments of companies and institutions, and at the same time, the achievement of such projects gives policy makers the confidence to demand higher energy standards [71].

The review also found that in the selected time-period, these countries share the same landscapes: The signing of the Paris Agreement for the mitigation of consequences climate change; and the EU directive on building's energy performance for the reduction of energy consumption and demands of buildings. Indeed, these actions put pressures on governments, which intensified the development and promotion of national strategies for reduce impact of buildings on the environment, hence mitigating climate change effects.

Table 1 summarizes regimes changes and landscape factors, sorted by country and in chronological order.

From the niche perspective, companies and institutions are implementing, through research and development activities, solutions to enhance buildings' features—such as energy efficiency, use of materials, etc.—in the Arctic. One example is the project *GrAB—Green Arctic Buildings*, which comprehends five institutes from four different countries: UiT The Arctic University of Norway (Norway); Murmansk State Technical University (Russia); Petrozavodsk State University (Russia); University of Oulu (Finland); Umeå University (Sweden). The aim of the project is to do research on sustainable building to enhance the region's competitiveness, improve life quality, and support social and economic activities with regard to environmental issues in the Arctic. The project is co-financed by the European Union; the Regional Council of Lapland; The Norwegian Kolartic; and the involved institutions.

Therefore, the identification of a niches, regimes, and landscape dimension leads to the conclusion that the transition process to sustainability in the building sector is taking place in the Arctic. However, as introduced in Section 3.1, despite that these measures are somehow contributing, for any of the discussed countries, national action plans directly involving the role of buildings in the region were found.

In fact, implementing strategies for the territory customized on Arctic climate conditions would have national impact. As shown in Table 2, arctic population accounts for the 12% of the total population in Finland; 9% for Norway; 5% for Sweden and Russia; 0.22% for USA; 0.11% for Canada. The same table reports that the size of the Arctic building stock is also relevant: 12% of buildings in Norway and Finland lie in the Arctic zones; 5% for Sweden; 0.27% for Canada; 0.23% for USA. For Iceland, whose total surface is falling into the Arctic boundaries, those percentages correspond to 100%.

**Table 1.** Summary of policies analysis.

| Country | Year | Regime Changes | Landscape Factors |
|---|---|---|---|
| Norway | 2009 | "Building for the future—environmental action plan for the housing and building sector 2009–2012" | |
| | 2016 | "New building blocks for the high north" "The property sector's Roadmap towards 2050" | |
| | 2017 | Building code update (TEK17) | |
| | 2017 | "Norway's Arctic Strategies between geopolitics and social development" | |
| | 2019 | "Think twice before demolishing" | |
| Sweden | 2008 | New energy labelling directive | |
| | 2010 | The Planning and Building Act Implementation of energy performance certificates for buildings | |
| | 2011 | The Planning and Building Ordinance | |
| | 2012 | Minimum energy performance standards | |
| | 2014 | "Sweden's strategy for the Arctic Region" | |
| | 2017 | "Sweden's Seventh Communication on Climate Change" | |
| | 2018 | "Sweden's draft integrated national energy and climate plan" | |
| Finland | 2012 | "Finland Strategies for the Arctic Region 2013" | Paris Agreement Update of the EU directive on the energy performance of buildings (2010/31/EU) |
| | 2013 | Update of National building code | |
| | 2014 | "Energy and Climate Roadmap 2050" | |
| | 2017 | Government Action Plan 2017–2019 | |
| Iceland | 2018 | "Iceland's Climate Action Plan for 2018–2030" | |
| | 2019 | "Iceland's Seventh National Communication and Third "Together Towards a Sustainable Arctic" | |
| Russia | 2008 | Basic Principles of Russian Federation State Policy in the Arctic to 2020 | |
| | 2016 | Implementation of building energy efficiency class standards Road map for EE building and structures | |
| | 2017 | Implementation of buildings envelope standards Implementation of building energy efficiency standard | |
| | 2020 | Update Arctic Policy | |
| Canada | 2013 | Implementation of net-zero energy building code | |
| | 2019 | Update of Arctic and Northern Policy | |
| | 2020 | Update of Zero Carbon Building standards V2 | |
| USA | 2016 | Update of national strategy for the Arctic region | |
| | 2018 | Update of building energy efficiency standards Climate Change Action Plan Recommendation to the Governor" | |

**Table 2.** Arctic countries population and number of buildings [72–77].

| Country | Population | Total Number of Buildings | Arctic Counties | Population | Total Number of Buildings | % Population in the Arctic | % Buildings in the Arctic |
|---|---|---|---|---|---|---|---|
| Norway | 5,374,807 | 4,212,721 | Nordland | 241,235 | 259,412 | 9.02% | 11.83% |
| | | | Troms and Finmark | 243,311 | 237,768 | | |
| | | | Svalbard | - | 1093 | | |
| Sweden | 10,343,403 | 4,978,239 | Vasterbotten | 272,044 | 141,434 | 5.05% | 5.54% |
| | | | Norrbotten | 249,843 | 134,142 | | |
| Finland | 5,533,390 | 2,734,219 | Northern Ostrobothnia | 412,830 | 192,183 | 11.97% | 11.59% |
| | | | Kainuu | 72,306 | 36,734 | | |
| | | | Lapland | 177,161 | 88,082 | | |
| Iceland | 366,700 | 90,494 | Iceland | 366,700 | 90,494 | 100% | 100% |
| Russia | 146,745,098 | n/a | Murmansk | 741,545 | n/a | 4.89% | - |
| | | | Nenets | 44,110 | n/a | | |
| | | | Yamal-Nenets | 544,008 | n/a | | |
| | | | Chukotka Autonomous Okrug | 50,726 | n/a | | |
| | | | Arkhangelsk | 1,136,387 | n/a | | |
| | | | Komi Republic | 820,171 | n/a | | |
| | | | Yakutia | 970,105 | n/a | | |
| Canada | 38,005,238 | 14,790,400 | Northwest Territories | 45,161 | 14,980 | 0.11% | 0.27% |
| | | | Nunavut | 39,535 | 9815 | | |
| | | | Yukon | 4205 | 15,215 | | |
| USA | 330,495,805 | 139,684,244 | Alaska | 731,545 | 319,854 | 0.22% | 0.23% |

Moreover, the partial data reported in Table 3 show a common pattern between Norway, Sweden and USA: The average total energy consumption per household per year in the Arctic is higher than national average. The same table reports that the average energy consumption in the Arctic Europe is lower than average energy consumption in Alaska. Between the Arctic countries, Canada is showing a different trend: The average energy consumption per households is lower than the national average. Among the selected countries, Norway has the lower national average energy consumption, followed by Canada, Sweden, Finland, and USA.

**Table 3.** Average total energy consumption by country [72–77].

| Country | Average Total Energy Consumption per Household [kWh] | Average Total Energy Consumption per Households in the Arctic Territories [kWh] |
|---|---|---|
| Norway | 20,230 | 23,056 |
| Sweden | 23,200 | 26,700 |
| Finland | 24,017 | n/a |
| Iceland | n/a | n/a |
| Russia | n/a | n/a |
| Canada | 28,237 | 15,626 |
| USA | 44,498 | 45,172 |

*4.2. Green Buildings Certification Systems*

The analysis points out inhomogeneity in the transition process among the discussed countries. In fact, results of transition can be identified by the size of the Green Buildings stock in the region. Table 4 collects, for each country, the total number of Green Building certifications released on national scale by the Green Building assessment tools introduced in Section 3.2. The table also includes the total number of Green Building certifications released in the Arctic, along with the name of the certified buildings and the achieved score. Data regarding certified buildings have been obtained from the official website of the rating system organizations.

The data presented reveal the low percentage of certified buildings in the Arctic, compared to the entire of each country. According to numbers provided in the table, certified buildings in the Arctic represents only the 1.59% of Green Buildings in Norway, 0.24% for Finland, 0.085% for Canada, 0.066% for Sweden, and 0% for Russia. Iceland, with the 100% of Green Building certified in the Arctic represents a special case because it has the whole area in the arctic region. Alaska represents 0.38% of Green Buildings in United States. Moreover, among the total 410 certified buildings, only 3 of them achieved the highest score (Platinum for LEED) and 104 with the second highest score (Gold for LEED and Excellent for BREEAM). The remaining 73% of buildings have been certified with an intermediate score (Pass, Good, or Very Good for BREEAM, and Silver or Certified for LEED). The reason for these results could be connected to the criteria evaluated over the certification process. In fact, despite the advanced technological level and economic possibilities of the examined countries allow buildings in the Arctic likely to satisfy each criteria and sub-criteria of any of the previously introduced rating system, some categories could penalize the overall evaluation. This is the case of the *Transport* category, in which BREEAM, LEED, or Green Zoom is emphasized. Indeed, this category evaluates the building by considering the proximity to amenities and facilities of the building site. Since the Arctic consists mainly of rural areas, construction of buildings can be penalized by the *Transport* criteria, and thus never achieve the maximum score. Also the transportation of materials to rural areas and the lack of a wide public transport system affects the evaluations of building in the Arctic. In the rating process, the *Energy* category could penalize the overall score because of the higher technological efforts required to achieve national standards in a higher energy demand area. Furthermore, as introduced in the Section 3.3, the Arctic is also less suitable for on-site renewable energy solutions.

**Table 4.** Certified Green Buildings in the Arctic region of Arctic Countries [78].

| Country | Rating System | Total Certifications | Arctic Certifications | Arctic Certified Buildings | Score |
|---|---|---|---|---|---|
| Norway | BREEAM-NOR | 304 | 4 | Office building Equinor—Harstad | 59.1% |
| | | | | Bodø 360—Bodø | 45.6% |
| | | | | Central Atrium—Bodø | 32.9% |
| | | | | Equinor building—Tromsø | 55.3% |
| | LEED | 9 | 1 | Building Aviation Authority—Bodø | Registered |
| Sweden | BREEAM-SE | 1174 | 0 | - | - |
| | LEED | 370 | 1 | Hotel Kiruna—Kiruna | SILVER |
| | Miljöbyggnad | n/a | n/a | - | - |
| Finland | BREEAM | 445 | 2 | Ramboll Finland Oy—Rovaniemi | 42.7% |
| | | | | Koy Tornio—Tornio | 52.1% |
| | LEED | 370 | 0 | - | - |
| | RTS GLT | n/a | n/a | - | - |
| Iceland | BREEAM | 10 | 10 | Höfdabakki 9—Reykjavik | 62.48% |
| | | | | Iceland Visitors Centre—Reykjavik | 53.9% |
| | | | | Icelandic Institute of Natural History—Garðabær | 48.8% |
| | | | | Upper Secondary School of Mosfellsbaer—Mosfellsbaer | 63.7% |
| | | | | Urridaholt—Gardabaer | 63.4% |
| | | | | Holmsheidi Prison—Reykjavik | 56% |
| | | | | Thingvellir National Park—Selfoss | 58.1% |
| | | | | Smáralind—Kopavogur (part 1—Asset Performance) | 57.4% |
| | | | | Smáralind—Kopavogur (part 2—Management Performance) | 63.1% |
| | | | | Sjúkrahótel (i.e., Patient Hotel)—Reykjavik | 81.09% |
| | LEED | n/a | n/a | - | - |
| Russia | BREEAM | 138 | 0 | - | - |
| | LEED | n/a | n/a | - | - |
| | GOST R 54954 | n/a | n/a | - | - |
| | Green Standard Certification System | n/a | n/a | - | - |
| | Green Zoom | n/a | n/a | - | - |
| Canada | LEED Canada | 5448 | 8 | Green Stone Building—Yellowknife | GOLD |
| | | | | Yellowknife Gallery Office Building—Yellowknife | SILVER |
| | | | | 38 & 40 Nijmegan Road—Whitehorse | GOLD |
| | | | | FH Collins Secondary School—Whitehorse | - |
| | | | | 704 Wood Street—Whitehorse | PLATINUM |
| | | | | 309 Main Street—Whitehorse | CERTIFIED |
| | | | | Whitehorse Hospital Staff Residence—Whitehorse | SILVER |
| | | | | IQALUIT International Airport Terminal Building | SILVER |
| | BOMA Best | 2260 | 0 | - | - |
| | LEED | 625 | 0 | - | - |
| | LEED for Homes | 872 | 0 | - | - |
| Alaska | LEED | 97,938 | 523 | - | CERTIFIED (18) SILVER (132) GOLD (101) PLATINUM (2) |
| | Energy Star | 36,498 | 0 | - | - |
| | Green Globes | 1632 | 0 | - | - |
| | Living Building Challenge | 98 | 0 | | |
| | National Green Building Standards | n/a | n/a | - | - |
| | WELL Building Standards | n/a | n/a | - | - |

Additional explanations could be related with cultural and economic factors. For instance, a study taking place in Russia [79] highlighted how people, during the certification process, are discouraged both by the technical regulations written in English, and by the high costs related to the certification process itself.

Future research opportunities should focus on the identification of factors encouraging and discouraging Green Buildings and upgrading of existing constructions to green standards in the Arctic, from an economic, social, and environmental point of view, along with the development of a rating tool customized on the characteristic features of the region.

### 4.3. Green Buildings Design Practices in Arctic Climate

Green Buildings design for Arctic climate conditions deals with different challenges compared to building design for a southern or continental climate. In fact, the achievement of environmental, economic, and social benefits—presented in Section 3.3—is possible only if the team responsible for the design takes into account the long and dark winters with low solar radiation and frozen ground, and the short, mild summers, characterizing the region.

Due to the low average temperatures, an efficient building envelope is one of the main important feature of green designing. This means minimizing heat losses and thermal bridges through a compact design and well-insulated building envelope. In particular, it is important to ensure that the structure is airtight, because cold air infiltrations can cause drafts and increase the need for heating. Moreover, it avoids moisture and condensation problems inside wall structure. Instead, a compact shape avoiding angles in the façade minimizes potential thermal bridges. By insulating the foundation slab from the building, it is also possible to avoid thermal bridges and minimize the risk of frost heave. Over the years, experimental solutions have also been tested. For instance, the study [80] proposes the installation of a single and a double glazing system on the south, east, and façade as energy saving solution for a building under renovation. A gently sloped roof covered with moss, sedum, or grass is also common design practice since it provides insulating effect and management control of storm water and snow.

A well-designed building envelope needs to be integrated with an efficient ventilation system. If also integrated with heat recovery, it can reduce energy demand, recovering heat from warmer indoor environments—such as the kitchen, laundry, and bathroom—and delivering it to colder environments—such as bedrooms and family rooms. The heat exchanger of the recovery unit should be designed on winter conditions, taking into account the minimum outdoor temperature and equipped with defrost function to avoid frosting under the extreme cold conditions.

As introduced in Section 3.3., renewable energy generation is a fundamental feature in Green Buildings design. Solutions for on-site domestic energy generation include solar panels and heat pumps. Even with the low solar radiation during winter, solar panels, if designed for winter conditions, can provide additional useful green energy, by exploiting additional energy from the light reflected by the snow [81]. Heat pumps are also feasible solutions, however a study on Net Zero-Energy Buildings in the Arctic Sweden [82], points out that air heat pumps can hardly achieve the Swedish energy requirements for Near Zero Energy Buildings. The same study identifies geothermal heat pumps and air-water heat pumps as better solutions.

Design practices also involve: Water recycling, to reduce energy demand; low consumption and efficient lighting systems, enhancing indoor thermal comfort during winter; and smart control systems, providing information on temperatures, water, and energy use to the users.

During the design process, material selection represents a significant feature for meeting green standards. Green materials should have the following characteristics: Recyclable; locally produced; energy-efficient; and with minimal impact on the environment and human health. In the Arctic, wood is a material fitting these requirements and therefore frequently used by the construction industry.

Finally, in view of climate change and the dramatic consequences on the Arctic environment, Green Buildings are asked to be adaptable to different case scenarios. In particular, the rising of the average temperature is requiring energy systems to also operate in off-design conditions and building

envelops to be adaptable to different range of temperatures. Moreover, the progressive melting of the permafrost is affecting the stability of building foundations, challenging structural engineers to reassess the current design and develop solutions that will ensure stable buildings over the years [83].

## 5. Conclusions and Future Research Opportunities

In this paper, a critical review of existing studies related to development of Green Buildings has been presented for seven of the Arctic countries. Even though there is an abundance of literature covering Green Buildings, the field is still lacking of studies specifically related to the Arctic. Below, the conclusions of each section are summarized:

- The review identifies the main actors in the Green Buildings development in the Arctic region. Indeed, global commitments aimed at mitigating climate change effects are leading governments to advance polices and national building codes responding to stricter standards, therefore pushing the construction industry and the market to adapt sustainable solutions. The transition to Green Buildings in the Arctic can be achieved if policies and building standards are implemented considering local climate and urbanistic patterns as well as the future local climate trend.
- Green Building rating tools are also playing a key role in promoting of sustainable, green constructions. Setting standards and requirements, they are pushing boundaries of sustainability in the building sector. The evaluation process takes into consideration different parameters according to different climate conditions and geographical characteristics, making the tools reliable and versatile. Despite the small number of buildings certificated in the Arctic, the criteria considered by the different tools showed the applicability of these systems in the Arctic. However, since Green Building rating tools are not designed on Arctic climate and local characteristics, the review identified that some of the evaluated criteria—such as transport and energy—are penalizing the achievement of certifications and high scores. In addition, more research is needed to identify the factors that are slowing the adoption of this type of sustainable solutions in the region.
- The review highlights general benefits and exposes criticalities of Green Buildings, focusing on the technologies needed for their development in the Arctic. In fact, the Arctic offers several solutions for green electricity generation. The challenge is creating a network that can reach rural areas, or alternatively, installing on-site production facilities. For this reason, it is necessary to develop technologies for on-site generation that can meet arctic requirements. To understand if Green Buildings located in the Arctic benefit from economic advantages, future research should also focus on arctic Green Building energy performance and cost analysis. In this way, it will be possible to calculate and estimate the average energy demand, energy savings, and the related accomplishments in economic terms.
- Green Buildings should be designed to meet current green standards and keep them over the time. Indeed, durability is an important feature of green design, and, since the Arctic is experiencing higher average temperature and permafrost melting, buildings should be designed taking into account these challenging changes in local climate patterns.

**Author Contributions:** Conceptualization, L.R. and R.R.; methodology, L.R. and R.R.; formal analysis, L.R.; investigation, L.R.; resources, L.R.; data curation, L.R.; writing—original draft preparation, L.R.; writing—review and editing, S.-E.S. and R.R.; supervision, S.-E.S. and R.R.; project administration, R.R. All authors have read and agreed to the published version of the manuscript.

**Funding:** This research received no external funding.

**Acknowledgments:** The publication charges for this article have been funded by a grant from the publication fund of UiT The Arctic University of Norway.

**Conflicts of Interest:** The authors declare no conflict of interest.

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
