# Peer review of "Green Building in the Arctic Region: State-of-the-Art and Future Research Opportunities"

_sustainability, doi:10.3390/su12229325_

Round 1
Reviewer 1 Report
Green Building in the Arctic Region: state-of-the-art and future research opportunities is presented.
This review provide the reader with a general overview of energy certification methods and polices for countries in the arctic region. The gap highlighted is stated as “Despite numerous studies and projects developed during the last decades a study describing the current research status for this region is still missing”. However, this paper fails to fully meet this objective. Instead this paper is mostly a collation of building energy polices and certification methods for the seven countries selected. There is no new insight provided.
For this paper to meet this stated objective the follow would be expected:
A meta-analysis for the selected countries which identifies difference in the performance of the building stock (based on energy efficiency and/or other performance metrics resulting from the ) building energy polices/regulation and certification present in each countries. I.e. which countries have the most efficient buildings and is it because of the specific policies or not.
What is the state of the art in building energy design and implementation in each country. Which country is leading the way in research within this field and how is it liked to government polices and regulations.
A comparison of the arctic region countries and other similar counties in northern Europe and northern continental USA. How do the policies and certifications methods in the Artic differ to these regions and what are the results? The Certification methods in the Artic are very similar to the regions already.
What is the future for building energy research and polices in the Artic region? With the significant increase in heating (especially in the summer months) in the Artic region, how will policy/research adapt? Would the adoption of methods from Northern European maritime climates be suitable? What about natural ventilation as a future zero carbon cooling option for the Arctic region?
Table 1 does not add anything.
Table 2 provides the BREEAM and LEED scores for the 7 countries, but no insight is provided on why other countries do better than other (apart from Iceland being completely in the Artic region). The author suggest some possible reasons such as decreasing population in the region an logistics issues and states that “this field clearly requires more investigation”. But what about the polices and research in these regions and their effect?
Reviewer 2 Report
Dear Authors
The artic region and green building is a challenging topic. The paper achieves the objectives, suggest:
Present and discuss in section 4.2. for buildings certify a summary of what are the solutions and technologies that are most adjust to the arctic region.
Small corrections and English revision: e.g line 12 CO2 must be CO2 , perhaps spoke of GHG Green House gas
Reviewer 3 Report
The paper is very well written and structured, and addresses a topic that is absolutely in line with the scope of this Journal. However, I find that it fails to address in an exhaustive way the aims emerging from the title (state-of-the-art of Green Buildings in Arctic regions) and from the Abstract (“This study provides a systematic literature review of existing research related to Green Buildings in the Arctic”).
What I mean is that, starting from these premises, I would have expected a paper including a discussion about how a Green Building must be conceived and designed in Arctic regions, with technical information regarding e.g. the materials to be used, the thickness of insulating materials, the heat recovery strategies, the wind-shielding strategies and so on. Instead, the paper includes just a general discussion about legislation, rating systems used in Arctic countries and general advantages of Green Buildings, where the peculiarities of the design in Arctic regions rarely emerge.
In my opinion, in order to be compliant with its scopes and to provide useful information for readers wanting to know more about Green Buildings in Arctic regions, the paper should address the following points:
- How large is the building stock in Arctic regions? What is the percentage buildings in Sweden, Norway, Finland, Russia, falling in the Arctic zone reported in Figure 1? I believe that such information is important to understand the potential benefit of building Green in Arctic regions: if Arctic regions just host a few buildings, the impact of Green Buildings in these areas on the overall energy/environmental balance of these Countries would be minor.
- How do the regulations valid for Arctic regions differ from those holding in the rest of the territory in a certain Country? Are there different technical specifications (e.g. thickness of insulation, maximum primary energy needs)? This would help readers to understand what peculiarities should be considered when designing a building in Arctic regions.
- Making some practical examples about Green Buildings designed/built in Arctic regions. How do they differ from other Green Buildings in the same Countries? What are the most frequent technical solutions emerging from this review?
Without addressing these points, I think that the aims of the paper would be missed. The authors should then reformulate them, but the paper would still lack practical interest.
Round 2
Reviewer 1 Report
Adequate changes have been implemented. However the quality of english in these changes is very poor. For example see the last bullet point in the conclusion : "Policies and building standards should continuously adapt to the change of world scenario and market needs, promoting sustainability but also local variety. To follow the quickly changes in climate patterns and the policies trend, Green Buildings should be designed to meet different requirements and deal with more vary climate related situations." This is riddled with grammatical errors.
Once the quality of English and sentence structuring has been upgraded to a publishable level, the paper may be considered for processing. I will leave the editor decide whether the English is up to scratch for this Journal or not.
Reviewer 3 Report
The authors have addressed all issues raised in my review. The paper can be accepted after the following few minor issues are checked.
line 574: "In particular, it is important to ensure that the structure IS airtight"
line 580: reference lacking []
line 594: reference lacking []
line 508: 0.11 % for Canada
